# Application of GWAS and mGWAS in Livestock and Poultry Breeding

**DOI:** 10.3390/ani14162382

**Published:** 2024-08-16

**Authors:** Jing Ren, Zhendong Gao, Ying Lu, Mengfei Li, Jieyun Hong, Jiao Wu, Dongwang Wu, Weidong Deng, Dongmei Xi, Yuqing Chong

**Affiliations:** 1Key Laboratory of Animal Genetics, Breeding and Reproduction in the Plateau Mountainous Region, Ministry of Education, Guizhou University, Guiyang 550025, China; renjing20000425@163.com; 2Yunnan Provincial Key Laboratory of Animal Nutrition and Feed, Faculty of Animal Science and Technology, Yunnan Agricultural University, Kunming 650201, China; zander_gao@163.com (Z.G.); : mfli_2000@163.com (M.L.); hongjieyun@163.com (J.H.); 15229238680@163.com (J.W.); danwey@163.com (D.W.); dengwd@ynau.edu.cn (W.D.)

**Keywords:** GWAS, mGWAS, livestock, poultry, breeding, candidate gene

## Abstract

**Simple Summary:**

This article first introduces the basic principles and analysis process of genome-wide association studies and metabolome genome-wide association studies, then summarizes the latest research progress regarding their applications in animal genetic breeding, and finally briefly compares and elaborates on the advantages and disadvantages of genome-wide association studies and metabolome genome-wide association studies. Given the current emphasis on the analysis of genome-wide association studies and metabolome genome-wide association studies as a focal point of scientific research, our aim in publishing this review is to provide a resource for fellow scientists to efficiently grasp the advancements within our domain.

**Abstract:**

In recent years, genome-wide association studies (GWAS) and metabolome genome-wide association studies (mGWAS) have emerged as crucial methods for investigating complex traits in animals and plants. These have played pivotal roles in research on livestock and poultry breeding, facilitating a deeper understanding of genetic diversity, the relationship between genes, and genetic bases in livestock and poultry. This article provides a review of the applications of GWAS and mGWAS in animal genetic breeding, aiming to offer reference and inspiration for relevant researchers, promote innovation in animal genetic improvement and breeding methods, and contribute to the sustainable development of animal husbandry.

## 1. Introduction

The genome-wide association study (GWAS) employs single nucleotide polymorphisms (SNPs) in animal genomes. SNPs serve as molecular genetic markers utilized for association analysis, with a target trait across the entire genome. This approach is instrumental in exploring complex genetic characteristics [1]. Currently, the molecular markers predominantly employed in GWAS analysis are SNP markers. This methodology relies on Linkage Disequilibrium (LD) research principles to investigate the association between genetic variation and traits within populations. It aims to identify candidate genes and Quantitative Trait Loci (QTL) crucial in determining phenotypes, thereby elucidating their molecular mechanisms [2]. Metabolome genome-wide association study (mGWAS) is a genome-wide association analysis centered on metabolite molecules. It assesses the relative contents of various metabolic molecules detected in tissues as phenotype values. This approach entails the batch and precise localization of relevant candidate genes; the identification of functional genes associated with metabolic physiology and phenotype; and the elucidation of molecular biochemical mechanisms underlying related metabolic pathways [3]. This approach can address the limitations of traditional GWAS and more precisely pinpoint genetic markers or candidate genes associated with important economic traits. For instance, in human research, mGWAS analysis of metabolites in whole blood has revealed numerous genetic loci influencing metabolite content, offering fresh perspectives for numerous complex diseases [4]. Several studies have been conducted on important crops and fruits such as rice [5], corn [6], wheat [7], tomatoes [8], and apples [9], providing vital references for exploring the genetic basis of plant metabolomics and advancing breeding efforts to enhance nutritional value. The integration and evolution of the mGWAS approach in livestock and poultry breeding stands as a significant indicator of the application of modern biotechnology in agricultural production. Here, we present a concise historical account of the introduction and development of the mGWAS method in livestock and poultry farming: In the late 20th and early 21st centuries, as genomics technology advanced, scientists employed GWAS to identify genetic variants linked to complex traits. However, traditional GWAS primarily focused on plants and human health [10,11], with limited applications in livestock and poultry farming. It was not until 2009 that researchers suggested metabolites could serve as biomarkers for livestock traits and diseases [12]. As metabolomics technology gradually matures, scientists are beginning to recognize the important bridging role of metabolites between phenotype and genotype [13]. With the accumulation of metabolomics data and enhancements in analytical tools, the mGWAS method was born. Initially, mGWAS was primarily applied to understanding the genetic bases of complex traits in plants [14]. As the technology continues to evolve, it unlocks new possibilities for both biological and medical research [15]. In 2020, for the first time, scientists employed mGWAS to analyze the genetic bases of plasma metabolites in hybrid beef cattle, aiming to estimate the genomic heritability of Canadian crossbred beef cattle [16]. Therefore, comprehending the genetic basis of animal metabolism is essential for enhancing animal breeds in livestock and animal husbandry. Both of these methods are extensively applied in the field of livestock and poultry breeding, holding significant importance for analyzing the genetic bases of livestock and poultry, improving their varieties, and promoting sustainable development. A flowchart of GWAS and mGWAS is illustrated in Figure 1. This review article presents the application and research progress of GWAS and mGWAS in animal genetic breeding, aiming to provide guidance for further in-depth exploration of these methods.

## 2. Research Progress of GWAS in Genetic Breeding of Livestock and Poultry

Livestock and poultry genetic breeding stands as a crucial component within the agricultural domain, holding significant importance for enhancing production performance, economic viability, and adaptation to environmental shifts. With the continual advancement of genomics technology, GWAS has found extensive application in genetic breeding research concerning cattle, sheep, pigs, and poultry, yielding numerous research outcomes. Table 1 provides a synopsis of the recent applications of GWAS technology in livestock and poultry studies.

In addition, there are also studies on the metabolic traits of livestock—Zhang et al. [48] studied the genetic correlation between the composition of longissimus muscle fatty acids and 32 traits related to growth, carcass, fat deposition, and meat quality in six different pig populations, including Bama Xiang pigs and Erhualian pigs. Most of the significant loci of saturated fatty acids (C14:0, C16:0, and C18:0) and monosaturated fatty acids (C18:1n9 and C16:1n7) identified using GWAS, including loci near the *ELOVL6*, *SCD*, and *FASN* genes, showed negligible or weak effects on all 32 physical traits. Backfat thickness and intramuscular fat content have a strong negative genetic correlation with C18:2n6, and a positive genetic correlation with C18:1n9. In all six populations, there was a consistent positive correlation observed between intramuscular fat content and saturated fatty acids. This observation sheds light on the shared genetic regulation of fatty acid composition alongside other economically relevant traits, thereby aiding in the enhancement of pork fatty acid composition from a genetic standpoint. In their 2016 study, Zhang and colleagues [49] performed a GWAS analysis on 33 fatty acid metabolism traits across five pig populations, identifying 865 SNPs and uncovering four new population-specific loci. This study demonstrated that focusing on intermediate phenotypes, such as fatty acid metabolism traits, enhances the statistical power of GWAS for end phenotypes. The study suggested three novel genes: *FADS2*, *SREBF1*, and *PLA2G7*, advancing our understanding of the genetic underpinnings of fatty acid composition in pigs. In Tetens’ study [50], 248 Holstein Friesian cows were genotyped, and genome-wide association studies were conducted on the levels of phosphocholine and glycerophosphocholine in their milk, as well as the ratio of these two metabolites. The results indicated that the latter two traits are heritable. A significant quantitative trait locus was identified on chromosome 25 of the cattle. The *APOBR* gene, which encodes the apolipoprotein B receptor, is located within this region and was analyzed as a candidate gene. The analysis revealed that polymorphisms within the gene are strongly associated with glycerophosphocholine levels and metabolite ratios. These findings indicate that variations in the ability to absorb blood phosphatidylcholine from low-density lipoprotein play a significant role in the metabolic stability of cows early in lactation, and suggest that *APOBR* harbors pathogenic variants.

Additional research on the economic traits of livestock includes a study conducted by Jun Park [51], where GWAS analysis was employed to identify the genomic regions and candidate genes associated with age to 105kg; average daily weight gain; backfat thickness; and eye muscle area in Yorkshire pigs. The results revealed significant chromosomal regions linked to 105 kg body weight age and average daily weight gain on the *Sus scrofa* chromosomes 1, 6, 8, and 16. Additionally, chromosomal regions significantly correlated with backfat thickness were found on chromosomes 2, 5, and 8 of *Sus scrofa*. Moreover, a chromosomal region significantly associated with eye muscle area was identified on chromosome 1 of *Sus scrofa*. These findings provide valuable insights for genetic improvement efforts in pigs. In their study on growth traits in pigs, Wang et al. [52] conducted a genome-wide association study based on site-specific amplified segment sequencing (SLAF-seq) and analyzed 10 growth traits in 223 four-way hybrid pigs. A total of 53 SNPs were identified for 10 growth traits. Thirteen candidate genes (*ATP5O*, *GHRHR*, *TRIM55*, *EIF2AK1*, *PLEKHA1*, *BRAP*, *COL11A2*, *HMGA1*, *NHLRC1*, *SGSM1*, *NFATC2*, *MAML1*, and *PSD3*) were found to be associated with growth traits in pigs. Wang et al. believe that these detected SNPs and the corresponding candidate genes may provide a biological basis for improving the growth and production performance of pigs. Reis et al. [53] used GWAS analysis to identify nine genomic regions related to the rib eye region and eight regions related to subcutaneous fat thickness in their study of Nellore cattle carcass and meat traits. Among them, the candidate gene *MTTP is* directly related to lipid metabolism, which may explain the changes in subcutaneous fat deposition in cattle; four regions are related to shear force, among which the candidate genes *HSP90B1* and *SCD5* are related to meat tenderness; three genomic regions related to intramuscular fat were identified; and four different genomic regions with candidate genes were identified on chromosomes 6, 8, 18, and 29. Among them, the *LONP2* candidate gene was the most prominent. This study provides important information on the genomic regions involved in analyzing traits, which helps improve beef quality. Velayudhan et al. [54] conducted GWAS analyses using Holstein Friesian cattle, Jersey cattle, and other breeds to identify candidate genes for daily milk production, non-fat solids, milk lactose, milk density, and clinical mastitis. They identified two SNPs significantly associated with daily milk production, rs109340659 and rs41571523, which are associated with two potential candidate genes, *FBRSL* and *CACN*; the GWAS identification of milk lactose identified SNPs close to the significance threshold (rs41634101), while no significant SNPs were detected for other traits. This study provides insights into the genetic association of milk production traits. Massender et al. [55] identified locations and functional candidate genes for milk yield and conformational traits in Canadian Alpine goat and Sashan sheep populations in a 2023 study. The GWAS identified 189 unique SNPs that are significant at the chromosomal level, and this study provides evidence that several candidate genes (e.g., *CSN1S1*, *CSN2*, *CSN1S2*, *CSN3*, *DGAT1*, *ZNF16*) are associated with milk composition traits in these populations. In addition, several new candidate genes (such as *DCK*, *MOB1B*, and *RPL8*) were proposed. Overall, their research provides insights into the genetic structures of economically important traits in Canadian dairy goat populations. Fu et al. [56] identified molecular markers and candidate genes related to egg-laying traits, and conducted whole genome sequencing analyses of Shuanglian chickens. Through whole genome sequencing and quality control, a total of 11,006,178 SNPs were obtained for further analysis. The GWAS results showed that 11 genome-wide significant SNPs and 23 suggestive significant SNPs are associated with egg production, maximum consecutive laying days, initial egg weight, and laying weight. Through functional annotation, three candidate genes, *NEO1*, *ADPGK*, and *CYP11A1*, were confirmed to be associated with egg production, while the *S1PR4*, *LDB2*, and *GRM8* genes are associated with maximum consecutive laying days, initial egg weight, and egg production weight, respectively. These findings may help us to better understand the molecular mechanisms of egg-laying traits in chickens and contribute to the genetic improvement of these traits.

In summary, the successful application of GWAS in livestock genetic breeding represents a significant advancement in genetic research and breeding practices. By screening loci and candidate genes associated with relevant traits, it offers improved breeding selection tools for stakeholders, thereby enhancing livestock and poultry traits such as growth, reproduction, and meat quality. Nonetheless, challenges persist in the interpretation and application of GWAS due to the complexity and diversity of livestock and poultry genomes. Hence, future research should focus on conducting more in-depth analyses of GWAS processes to further facilitate its application in the livestock and poultry industries.

## 3. Research Progress of mGWAS in Genetic Breeding of Livestock and Poultry

mGWAS analysis was initially mainly applied in the screening of molecular markers for human disease diagnosis and treatment [57]. By using it to study the metabolic products and genetic variations of organisms, we can explore the associations between metabolites and genetic variations such as genes, thereby revealing the interactions between metabolites and genetic factors. This comprehensive analysis of the genetic bases of complex traits further elucidates the functional relationships and regulatory mechanisms between genes and phenotypes, providing new insights and methods for selecting and breeding superior genes. In recent years, there have been numerous applications of mGWAS in the study of the genetic bases of crop metabolomics. For instance, Zhang et al. conducted an mGWAS analysis on buckwheat, providing a wide range of genomic resources for a deeper understanding of its origin, spread, and domestication. This will enhance future breeding in terms of trait improvement, especially in terms of yield and quality [58]. Unfortunately, as a newly emerging analytical method, there have been limited reports on the application of mGWAS in animal research. Currently, mGWAS research on cattle, pigs, and poultry is limited. Through mGWAS, we analyze plasma metabolites and other tissues to identify candidate genes beneficial for economic and metabolic traits. This enhances our understanding of the biological functions and processes influencing advantageous traits, thereby enhancing livestock genetic breeding. The following section highlights the various applications of mGWAS in animals.

The existing research on cattle using mGWAS is as follows: Li et al. [16] estimated the heritability of 33 plasma metabolites in Canadian hybrid beef cattle and found that the heritability of 11 metabolites was low to moderate, providing evidence for the genetic basis of metabolite concentration changes. Three significant SNP associations were detected for betaine (rs109862186), L-alanine (rs81117935), and L-lactate (rs42009425), indicating that genetic effects may be largely polygenic. The SNP of L-alanine was found to be located within the *CTNNA2* gene, which may be related to the regulation of L-alanine concentration in bovine blood. The estimated heritability values and candidate genes observed in this study will serve as an information resource for further research on the genetic improvement of hybrid beef cattle using plasma metabolites. In a 2021 study by Li and colleagues [59] focusing on beef cattle feed traits, genes and biological functions linked to residual feed intake were examined using inferred whole-genome DNA variations and 31 plasma metabolites, including their constituent traits. Through regression analyses of feed efficiency traits, metabolites, and mGWAS, several candidate genes were identified as being associated with residual feed intake and its constituent traits, including *PLSCR1*, *AQP9*, *NEDD4*, *PRTG*, *PYGO1*, *CUX2*, and *NOS1*. These findings enhance our understanding of the biochemical mechanisms behind feed efficiency traits and offer potential for improving the accuracy of genomic predictions through metabolite data integration. The following year, in their study exploring the genetic mechanisms underlying beef cattle carcass quality traits, Li et al. [60] combined regression analyses of metabolites and carcass quality traits with mGWAS results, identifying 103, 160, 83, 43, and 109 candidate genes, respectively, linked to hot carcass weight, rib eye area, average backfat thickness, lean meat yield, and carcass marbling score. In the case of the metabolites, several of the candidate genes identified via mGWAS are correlated with diverse carcass traits. For instance, *CDH13* is correlated with hot carcass weight, rib eye area, average backfat thickness, and carcass marbling score, while *KMT5B*, *NDUFS8*, *ALDH3B1*, *CHKA*, and *TCIRG1* are associated with hot carcass weight, rib eye area, lean meat yield, and carcass marbling score. This research advances our comprehension of the molecular and biological functions and processes that influence the advantages of carcass traits. Deng et al. [61] employed a multi-omics approach to analyze the genetic and metabolic determinants of milk traits in 100 water buffalo. Through mGWAS analysis, they identified 13 significant genetic variations, predominantly focused on l-proline. Notably, a single nucleotide polymorphism within the *ATG16L1* gene was found to be associated with proline production. These findings have significantly advanced our understanding of the genetic bases underlying metabolic traits in water buffalo and their milk.

mGWAS has also been used in pig research—in a study conducted by Wang [62] in 2020, mGWAS analysis was performed on 59 Duroc pigs and 49 Landrace pigs, with the aim of enhancing feed efficiency and elucidating the biochemical mechanisms associated with genetic variations in pig feed efficiency. The study identified 152 significant genome-wide SNPs that were linked to 17 metabolites, with 90 important SNPs annotated to 52 genes. Among these, SNPs located upstream of the *LRRC4C* gene exhibited a significant correlation with the metabolism of various glycerophospholipid molecules. The *SH2D4A* gene was found to influence intramuscular fat deposition by regulating triglyceride metabolism, while SNPs located in the intron and downstream region of the *MBOAT1* gene displayed a significant correlation with the synthesis of two types of monogalactosyl diacylglycerol.

The use of mGWAS in poultry research is as follows: in a study by Liu [63], metabolomics methods were employed to analyze 3431 metabolites and 702 volatiles in 423 skeletal muscle samples obtained from a gradient consanguinity segregating population resulting from hybridization between Beijing ducks and Liancheng ducks. A total of 2862 mGWAS signals were identified based on metabolomics, along with 48 candidate genes potentially regulating metabolite and volatility levels. The levels of 2-pyrrolidone in both the hydrophilic metabolome and volatile groups showed a significant correlation with QTL (4.56–5.23 Mbp) on chromosome 7. Furthermore, the expression of the *AOX1* gene was significantly associated with 2-pyrrolidone levels, suggesting that higher expression of *AOX1* in Liancheng ducks leads to elevated 2-pyrrolidone levels. These findings suggest that 2-pyrrolidone exhibits high stability and can serve as a direct molecular marker for metabolomics-assisted flavor breeding. Tian et al. [64] developed advanced hybrid lines by crossing the high-quality chicken line A03 with a local Chinese breed, the Huiyang Bearded chicken. By mGWAS, they successfully identified regulatory loci influencing metabolites and uncovered two candidate genes, *TDH* and *AASS*, associated with amino acid metabolites, as well as two other candidate genes, *ABCB1* and *CD36*, linked to lipids. This comprehensive investigation reported mGWAS signals for 253 metabolites, representing the most extensive study on the associations of chicken blood metabolites to date. The application of mGWAS in analyzing the genetic bases of chicken metabolic traits and metabolites holds potential for enhancing chicken breeding practices.

Unfortunately, we found no uses of mGWAS in studies on sheep. Table 2 summarizes the research progress mentioned above.

In summary, mGWAS demonstrates significant application value in the field of livestock breeding by effectively identifying genes associated with complex traits through related metabolites, yielding refined and accurate research results. Metabolomics has played a crucial role in animal research; however, the functions of certain metabolites remain unclear and may impact the outcomes of mGWAS analysis. Therefore, further functional annotation and genomic research are essential to unveil their potential biological mechanisms. Given the relatively complex analysis process of mGWAS, which necessitates a large sample size and relatively high research costs compared to GWAS, there have been limited achievements using mGWAS in the research of livestock and poultry. Nevertheless, we anticipate more innovative discoveries and breakthroughs in mGWAS usage for livestock breeding in the future.

## 4. The Advantages and Disadvantages of GWAS

Compared to traditional genetic research methods, GWAS offers the advantage of accurately pinpointing mutation sites and directly utilizing SNP/LD identification within genes or at the individual level (without lineage specificity) to identify new single genes and oligogenic disease genes. This approach unveils new biological mechanisms and provides insights into racial variations of complex traits [65]. Its primary benefits include enhanced flux analysis, statistical power, time savings, and reduced sequencing costs. GWAS enables the simultaneous association analysis of tens of thousands of SNPs or mutation sites, facilitating the increased detection of information that influences trait variation. Consequently, it is particularly suitable for investigating the genetic mechanisms underlying complex traits [66]. Moreover, GWAS data can be shared through networks and other means, extending its utility beyond gene recognition. Despite the aforementioned advantages of GWAS, it is important to acknowledge its shortcomings: (1) Due to the diverse genetic backgrounds of research subjects, the occurrence of stratification phenomena is likely. Consequently, it is imperative to hierarchically regulate the population structure to prevent false correlations stemming from it, which may lead to unreliable results [67]. (2) Research results obtained from different groups lack strong repeatability, and the heterogeneity of SNP effects complicates the replication of GWAS tests. Furthermore, significant differences exist in the detection results of the same phenotypic trait within the same research subjects among different research teams [68,69]. (3) The significant loci detected by GWAS analysis based on SNP markers can only account for a small fraction of the genetic variation in the phenotype, necessitating heightened attention to other forms of genomic variation [70]. (4) GWAS employs a genome point-by-point scanning method for detection, which limits the simultaneous utilization of regulatory information from multi-gene networks that control traits, posing challenges in detecting QTLs with moderate or low effects [2]. (5) GWAS solely utilizes genetic marker information at the DNA level, making it challenging to extract additional biological information at the molecular level. (6) Multiple hypothesis testing may lead to the exclusion of genetic variations with minimal impact on the phenotype, increasing the likelihood of false negative results. Hence, a considerable number of genetic variations across the entire genome may remain undetected [3].

## 5. The Advantages and Disadvantages of mGWAS 

Compared to conventional GWAS, metabolomics data provide more refined and accurate information, offering new insights into the genetic basis of metabolite diversity, particularly rare variations. Multidimensional omics research based on mGWAS facilitates the exploration of genetic diversity and the regulatory mechanisms of metabolites. In comparison to traditional phenotype GWAS, mGWAS allows for comprehensive metabolite analysis with higher accuracy. Given the significant variations in metabolite types and contents among different species and individuals, mGWAS technology can precisely identify candidate genes, thereby uncovering functional genes related to metabolic regulation [71]. However, mGWAS has certain limitations. Firstly, due to the intricate and interconnected metabolic pathways involved in metabolomics, the analysis process of mGWAS is relatively complex. Secondly, conducting mGWAS necessitates a large sample size, which may be constrained by practical limitations such as time availability. Lastly, mGWAS requires more advanced technology and equipment, resulting in higher research costs. Table 3 provides a brief comparison between mGWAS and GWAS.

## 6. Conclusions and Perspective

Metabolomics serves as a bridge between the genome and phenotype, representing the downstream of transcriptomics and proteomics. The actual size of a metabolome cannot be inferred from known genomic information using central rules, as the transcriptome or proteome may not align. Therefore, the integration of metabolomics and genomics can yield more valuable information [72]. With the in-depth exploration of livestock and poultry genomes, it is anticipated that GWAS and mGWAS analysis methods will become more refined and efficient, leading to the discovery of additional genes related to complex traits in the future. The application of GWAS and mGWAS technology can comprehensively elucidate the genetic principles underlying complex traits in livestock and poultry genetics and breeding, thereby expanding the scope for genetic enhancement. Moreover, as technology continues to advance, the accumulation and sharing of population and duplicate sample data is expected to accelerate genetic breeding research in livestock and poultry, as well as facilitate the translation and application of GWAS and mGWAS research outcomes. Consequently, the future development trends of GWAS and mGWAS in animal genetic breeding are poised to offer enhanced potential in technology, resource sharing, theoretical innovation, and other facets, providing more comprehensive support for animal genetic breeding research. Future research can further deepen the understanding of GWAS and mGWAS applications in animal genetic breeding through interdisciplinary collaboration; the development of novel data analysis techniques and bioinformatics tools; and the provision of more comprehensive and effective methods and strategies for animal genetic improvement, to enhance breeding efficiency, improve the quality of livestock and poultry products, and advance the development of precision animal husbandry. In summary, the introduction and development of mGWAS methods have brought new research perspectives and technological means to livestock and poultry breeding, promoting the scientific and precise management of modern animal husbandry.

## Figures and Tables

**Figure 1 animals-14-02382-f001:**
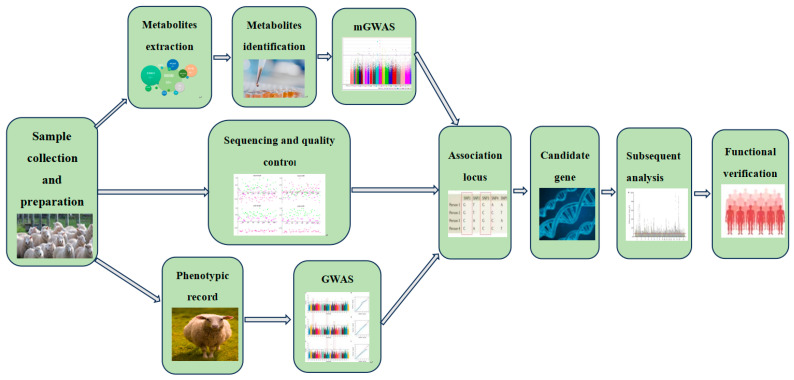
The flowchart of GWAS and mGWAS.

**Table 1 animals-14-02382-t001:** Application of GWAS Technology in Livestock and Poultry Research.

Species	Variety	Research Traits	Candidate Gene	References
Cattle	Indonesian cattle	Body weight	*SUGT1*, *SF3A3*, *DSCAM*	[17]
	Canadian Holstein cattle	Reproduction	*CSh*, *FSTCc*, *NRRh*	[18]
	Qinchuan cattle	Body conformation	*ADAMTS17*, *ALDH1A3*, *CHSY1*, *MAGEL2*, *MEF2A*, *SYNM*, *CNTNAP5*, *CTNNA3*	[19]
	Holstein cattle	Heifer livability	*MOG*, *OR12D2E*, *OR12D3*, *OR2H1*, *OR5V1*, *OR5V1C*, *OR5V2*, *TRIM10*, *TRIM15*	[20]
	Hawaiian beef cattle	Carcass weight	*RGS20*, *TCEA1*, *LYPLA1*, *MRPL15*, *EIF5*	[21]
	Brazilian beef cattle	Carcass and meat quality traits	*CAST*, *PLAG1*, *XKR4*, *PLAGL2*, *AQP3/AQP7*, *MYLK2*, *WWOX*, *CARTPT*, *PLA2G*, etc.	[22]
	Simmental Holstein cattle	Hair color and birth weight	*RNF41*, *ZC3H10*, *ERBB3*, *PMEL*, *OR10A7*	[23]
Sheep	Spanish Merino sheep	Quality wool	*EDN2*, *COL18A1*, *LRP1B*, *FGF12*, *ADAM17*	[24]
	U.S. rangeland ewes	Longevity and reproduction	*LPL*, *ANOS1*, *ARHGEF26*, *ASIC2*, *ASTN2*, *ATP8A2*, *CAMK2D*, etc.	[25]
	High mountain Merino sheep	14 months live weight	*FAM184B*, *NCAPG*, *MACF1*, *ANKRD44*, *DCAF16*, *FUK*, *LCORL*, *SYN3*	[26]
	Hu sheep	Shape	*KITLG*, *CADM2*, *MCTP1*, *COL4A6*	[27]
	Merino sheep	Fiber and skin wrinkles	*ALX4*, *EIF2AK2*, *ESRP1*, *HAS2*, *MC5R*, *MX2*	[28]
	5 varieties including Wadi, Icelandic, Finnsheep, etc.	Litter size	*CASK*, *PLCB4*, *RPTOR*, *GRIA2*, *PLCB1*	[29]
	Colombian Creole hair sheep	Meat quality	*ELOVL2*, *ARAP2*, *IBN2*, *TPM2*, etc.	[30]
Pig	Jinhua × Piétrain	Meat color	*ZBTB17*, *FAM131C*, *KIFC3*, *NTPCR*, etc.	[31]
	Large White × Tongcheng pigs	Intramuscular fat traits in longissimus dorsi muscle	*NR2F2*, *MCTP2*, *MTLN*, *ST3GAL5*, *NDUFAB1*, etc.	[32]
	Duroc × (Landrace × Yorkshire) pigs	Somatic skeletal traits	*OPRM1*, *SLC44A5*, *WASHC4*, *NOPCHAP1*, *RHOT1*, etc.	[33]
	Yorkshire pigs	Reproductive traits	*ELMO1*, *AOAH*, *INSIG2*, *NUP205*, *LYPLAL1*, etc.	[34]
	Duroc, Changbai, Dabai	Growth traits	*SKAP2*, *SATB1*, *PDE7B*, *PPP1R16B*, *WNT3*, *WNT9B*	[35]
	Duroc × Landrace × Yorkshire	Economic characteristics of carcass	*TIMP2*, *EML1*, *SMN1*	[36]
	Duroc × Saba, Yorkshire × (Landrace × Saba)	Meat quality traits	*GRM8*, *ANKRD6*, *MACROD2*, *CDYL2*, *CHL1*, etc.	[37]
	Duroc, Yorkshire, Landrace	Pig fatness trait	*MC4R*, *PPARD*, *SLC27A1*, *PHLPP1*, *NUDT3*, *ILRUN*, *RELCH*, *KCNQ5*, *ITPR3*, and *U3*	[38]
Birds	Wenchang chickens	Feed efficiency and growth traits	*PLCE1*, *LAP3*, *MED28*, *QDPR*, *LDB2* and *SEL1L3*	[39]
	Italian local chickens	Shank and eggshell color	*MTAP*, *CDKN2A*, *CDKN2B*, *SLC7A11* and *MITF*	[40]
	Wenshang Barred, Recessive White, Luxi Mini	Body weight and size	*LCORL*, *LDB2*, and *PPARGC1A*	[41]
	Rhode Island Red chickens	Eggshell strength	*FRY* and *PCNX2*	[42]
	Ogye x White Leghorn	Skin color	*MTAP*, *FEM1C*, *GNAS* and *EDN3*	[43]
	Lingnan Yellow chicken×Chinese Huiyang Bearded	Body weight	*CAB39L*, *RCBTB1*	[44]
	Arbor Acres broiler× Baier layer	Skeletal muscle production traits	*LRCH1*, *CDADC1*, *CAB39L*, *FOXO1*, *NBEA*, *GPALPP1*, etc.	[45]
	Tibetan chicken, Wenchang chicken, etc.	Chest muscle fatty acid composition	*ENO1*, *ADH1*, *ASAH1*, *ADH1C*, *PIK3CD*, *WISP1*, *AKT1*, *PANK3*, *C1QTNF2*	[46]
	NEAUHLF	Growth traits	*ACTA1*, *IGF2BP1*, *TAPT1*, *LDB2*, *PRKCA*, *TGFBR2*, *GLI3*, *SLC16A7*, *INHBA*, *BAMBI*, *GATA4*, etc.	[47]

**Table 2 animals-14-02382-t002:** Application of mGWAS Technology in Livestock and Poultry Research.

Species	Variety	Research Contents	Candidate Gene	References
Cattle	Canadian hybrid beef cattle	Heritability of plasma metabolites	*CTNNA2*	[16]
	Charolais, Hereford–Angus crosses, and a Beefbooster composite breed	Feed efficiency traits	*PLSCR1*, *AQP9*, *NEDD4*, *PRTG*, *PYGO1*, *CUX2*, *NOS1*, etc.	[59]
	Charolais, Hereford–Angus crosses, and a Beefbooster composite breed	Carcass merit traits	*CDH13*, *KMT5B*, *NDUFS8*, *ALDH3B1*, *CHKA*, *TCIRG1*	[60]
	Guangxi Buffalo	Buffalo milk traits	*ATG16L1*	[61]
Pigs	Duroc pig, Landrace pig	Genetic variation in feed efficiency	*LRRC4C*, *SH2D4A*, *MBOAT1*	[62]
Birds	Beijing Duck × Liancheng Duck	Skeletal muscle metabolism	*AOX1*, *ACBD5*, *GADL1*, *CARNMT2*	[63]
	High-quality chicken strain A03 × Huiyang Bearded Chicken	Chicken blood metabolites from the Qing Dynasty	*TDH*, *AASS*, *ABCB1*, *CD36*	[64]

**Table 3 animals-14-02382-t003:** Comparative Summary of GWAS and mGWAS.

	mGWAS	GWAS
Principle	Metabolite genome association analysis is conducted, utilizing sample metabolomic data as the phenotype [72].	Phenotypic trait information was gathered from samples, followed by an association analysis between the phenotype and the genome [1].
Characteristic	Among different varieties or individuals, the types and contents of metabolites exhibit significant variation, characterized by rich data and precise identification.	Traditional phenotypes exhibit fewer types, are challenging to quantify, and are heavily influenced by environmental factors, leading to elevated rates of false positives.
Summary	The increased availability of phenotypic data enables the identification of a broader spectrum of gene phenotypes that can be quantified. Enhanced quantification accuracy corresponds to more precise SNP localization. Moreover, ample data facilitate the identification of rare SNP loci.	The localization of genes is relatively limited in terms of quantity, and their localization effect is often poor. Additionally, there can be simultaneous associations with multiple genes, making it challenging to distinguish the main gene responsible.

## Data Availability

Not applicable.

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
