# Peer review of "Application of GWAS and mGWAS in Livestock and Poultry Breeding"

_animals, 2024, doi:10.3390/ani14162382_

Round 1

Reviewer 1 Report

Comments and Suggestions for Authors

Ren and colleagues summarizes the advance of GWAS and mGWAS research in livestock and poultry breeding. The topic is interesting. However, the description of the advances has some problems which in my eyes are necessary to reliably assess its quality prior to publication.

1. I would suggest a change in the title of the article to Application of GWAS and mGWAS in Livestock and Poultry Breeding

2. In part 2 and part, it appears that the authors only addressed the instances listed, I suggest the author to categorise descriptions according to species or traits.  This may make the advance clearer to the reader.

3. It is better the introduce the metabolic traits and related economic traits in livestock and poultry.

4. Some future research directions are suggested in the outlook.

5. The manuscript still needs to be proofread carefully for English grammar errors. longest back muscle and longest muscle is not correct.

Comments on the Quality of English Language

The manuscript still needs to be proofread carefully for English grammar errors.

Reviewer 2 Report

Comments and Suggestions for Authors

This interesting article provides an overview of the use of two powerful tools for analyzing genomic variants associated with phenotypes, allowing for in-depth exploration of livestock and poultry genomes. Indeed, genome-wide association studies (GWAS) have greatly aided in deciphering significant genetic loci that account for economic impact. And it’s well-known that GWAS has been guiding the livestock and poultry industries for the past 20 years, and the release of metabolomics-genome association studies (mGWAS) has already led to the discovery of additional genes associated with complex genes.

The current paper is characterized by the timeliness, the breadth of the discussion and may provide guidance for further research. The manuscript is well structured and most of the references refer to the studies from this year or the last 2–5 years.

The only thing I would recommend to the authors and what I really missed in it –it is a tracing of the very history of the introduction of the  above-mentioned methods in livestock and poultry farming. I am aware that the authors have concentrated their attention on the presentation of the application and research progress of GWAS and mGWAS in animal genetic breeding. Therefore, this is not a remark, it’s only  a recommendation, taking into account which can make the article even more interesting for a scientific audience. In this regard, it remains at the discretion of the authors themselves.
